# Progress on Phage Display Technology: Tailoring Antibodies for Cancer Immunotherapy

**DOI:** 10.3390/v15091903

**Published:** 2023-09-09

**Authors:** Renato Kaylan Alves França, Igor Cabral Studart, Marcus Rafael Lobo Bezerra, Larissa Queiroz Pontes, Antonio Marcos Aires Barbosa, Marcelo Macedo Brigido, Gilvan Pessoa Furtado, Andréa Queiroz Maranhão

**Affiliations:** 1Molecular Immunology Laboratory, Department of Cellular Biology, Institute of Biological Sciences, University of Brasilia, Brasilia 70910-900, Brazil; 200086251@aluno.unb.br (R.K.A.F.); brigido@unb.br (M.M.B.); 2Graduate Program in Molecular Pathology, University of Brasilia, Brasilia 70910-900, Brazil; 3Oswaldo Cruz Foundation, Fiocruz Ceará, Eusébio 61773-270, Brazil; igor.studart@fiocruz.br (I.C.S.); marcus.lobo@fiocruz.br (M.R.L.B.); larissa.pontes@fiocruz.br (L.Q.P.); marcos.aires@fiocruz.br (A.M.A.B.); gilvan.furtado@fiocruz.br (G.P.F.); 4Graduate Program in Biotechnology of Natural Resources, Federal University of Ceará, Fortaleza 60440-970, Brazil; 5Graduate Program in Applied Informatics, University of Fortaleza, Fortaleza 60811-905, Brazil

**Keywords:** Phage Display, cancer therapy, therapeutic antibody, biopanning

## Abstract

The search for innovative anti-cancer drugs remains a challenge. Over the past three decades, antibodies have emerged as an essential asset in successful cancer therapy. The major obstacle in developing anti-cancer antibodies is the need for non-immunogenic antibodies against human antigens. This unique requirement highlights a disadvantage to using traditional hybridoma technology and thus demands alternative approaches, such as humanizing murine monoclonal antibodies. To overcome these hurdles, human monoclonal antibodies can be obtained directly from Phage Display libraries, a groundbreaking tool for antibody selection. These libraries consist of genetically engineered viruses, or phages, which can exhibit antibody fragments, such as scFv or Fab on their capsid. This innovation allows the in vitro selection of novel molecules directed towards cancer antigens. As foreseen when Phage Display was first described, nowadays, several Phage Display-derived antibodies have entered clinical settings or are undergoing clinical evaluation. This comprehensive review unveils the remarkable progress in this field and the possibilities of using clever strategies for phage selection and tailoring the refinement of antibodies aimed at increasingly specific targets. Moreover, the use of selected antibodies in cutting-edge formats is discussed, such as CAR (chimeric antigen receptor) in CAR T-cell therapy or ADC (antibody drug conjugate), amplifying the spectrum of potential therapeutic avenues.

## 1. Introduction

Phage Display technology is a method to present proteins, such as antibody fragments, on the surface of filamentous phages. George Smith first demonstrated in 1985 that it is possible to display exogenous proteins fused to the filamentous phage f1 capsid, while maintaining the viral infectivity and protein’s biological activity [1]. Achieving protein display on the surface of a phage requires a strategic manipulation of its genome. This involves the fusion of a gene encoding the desired protein to the 5′ region of a filamentous phage’s coat protein, such as protein III or protein VIII. Once the phage particle is assembled, the exogenous protein is expressed as part of the viral capsid [2].

The methodology evolved to allow the construction of libraries of polypeptides displayed on the phage surface. As the displayed proteins are exposed in a manner that allows their interaction with ligands, the libraries can be used to select polypeptides with affinity to a particular ligand. This approach proves valuable in investigations concerning protein–protein interactions, the definition of epitopes, or even the search for enzyme substrates [3]. For example, it is possible to build high-variability libraries to search for an antibody with high affinity to a given antigen. Each phage works as a fusion unit and as an independent vector, so a particle with affinity to the specific ligand can be identified/selected, and this particle can separately be propagated [4].

The selection of the Phage Display involves searching for antibodies with specificity to an antigen of interest, in a vast antibody ocean that consist of the libraries (immune, non-immune or synthetic) of antibodies [5]. Historically, a metaphor has been used that associates this selection with panning—the work of a miner searching an ore material in search of gold—or a metaphor in which a fisherman would fish for a particular antibody in an ocean of great diversity of antibodies. In both analogies, the idea is that, in theory, an antibody can be isolated against any antigen, whether it is immune or not, or even against a toxic antigen [6].

Several antibodies approved for therapeutic use have been developed using Phage Display (Table 1), which is an alternative to the hybridoma technology and enables the discovery and generation of fully human antibodies against specific targets [7]. Antibodies, also known as immunoglobulins, comprise structures characterized by heavy chains containing constant domains (CH) and a variable domain (VH), and light chains, containing a constant domain (CL) and a light domain (VL) [8]. An immunoglobulin G is constituted by a symmetrical arrangement of two identical heavy chains and two identical light chains [9]. There are different classes of antibodies, and in addition to the natural forms, it is possible to construct antibody fragments containing only some parts of the immunoglobulin molecule [10]. In Phage Display libraries, antibody fragments such as single-chain fragment variable (scFv), containing only VH and VL interconnected by a flexible linker peptide, antigen binding fragments (Fab), containing VH, CH1, VL and CL, and single domain, mainly VHH (camelid-derived heavy chain variable domain), are usually displayed. The reduced size of these fragments enables their display surface without interfering with the particle assembly or infectivity [11].

Therapeutic monoclonal antibodies (mAbs) and their derivatives have revolutionized oncology treatment by leveraging the specificity observed in natural effector mechanisms of the adaptive immune system [12]. These mAbs, whether used as naked molecules, conjugated to cytotoxic payloads, or incorporated into chimeric antigen receptors (CARs), have enabled targeted and precise treatments in diverse oncological scenarios. Ongoing advancements in this field continue to shape the landscape of cancer therapy [13]. Phage Display has assumed a significant role in advancing anti-cancer strategies by facilitating the discovery of novel antibodies, antigens, and epitopes associated with tumors. Additionally, it enables various strategies to tailor affinity, reduce immunogenicity, and search for biophysical properties that enhance the druggability of these therapeutic molecules [14].

While Phage Display-based antibody discovery platforms can employ repertoires obtained from immunized animals or patients with particular tumors to enable the in vitro selection of molecules with desired characteristics, a key strength of this methodology relies on its independence from animal (or patient) involvement for generating fully human therapeutic mAbs [2].

This review provides an overview of the impact of Phage Display in antibody research, in general, and specifically for tumors, presenting the antibodies approved for therapeutic use developed using this approach and putting forward a strategy of antibody development for the treatment of different types of cancer. Furthermore, we elucidate the primary selection methodologies and categories of antibody libraries that have been employed.

## 2. Main Types of Phage Display Libraries Used to Select Antibodies

Phage Display has been widely applied to develop novel therapeutic antibodies. Impressively, 18% of all antibodies commercially approved owe their origins to this technology. Most of the Phage Display-derived antibodies that have gained clinical approval have been tailored for deployment in cancer (36%) and immunoregulation (32%) applications (Figure 1A) (Table 1).

Since the early 1990s, when Phage Display started to become a popular strategy to select antibody fragments, several types of libraries have been developed. One of the first libraries constructed [15] used mouse scFv and Fab fragments. In the same year, Barbas III and co-workers (1991) developed a protocol to construct combinatorial libraries from human repertoire based on Fab fragments [16]. These libraries rely on the random rearrangement of heavy and light chains amplified from immune cells, either from immunized or naïve animals. Zebedee et al. (1992) used a library to select Fab fragments from a vaccinated individual, which can be considered an immune library from a human subject. Libraries derived from naturally immunized individuals, such as vaccinated subjects, are difficult to find, but even non-immunized human libraries have been used successfully to obtain high-affinity antibodies against either self- or non-self-antigens [17].

Phage Display libraries can also be developed from variants of a single antibody. One of these approaches relies on the use of directed evolution to select high-affinity mutants from a scFv library that has been randomly mutated through error-prone PCR (epPCR), as previously described [18,19]. Random mutagenesis might be useful to obtain mature antibodies with different kinetic parameters (different association/dissociation rates). Direct evolution of antibody affinity can also be accomplished using libraries of CDRs (complementary determinant regions) of variable domains, mainly CDR3H. CDR3H variants were randomly generated using degenerate oligonucleotides where codons such as NNS were used, raising the possibility of any of the twenty amino acids being coded, while reducing the change to an unwanted stop-codon [20].

Antibody fragments may also be isolated from infected organisms [21,22,23], as demonstrated by selecting Fab fragments from HIV-1-positive individuals. This is particularly interesting for infectious diseases as these subjects are considered immunized and, therefore, produce higher--affinity antibodies. This strategy has also been previously described to select IgE antibodies from an atopic donor immunized with tetanus toxoid [24]. Other examples are an anti-U1 RNA-binding antibody fragment [25] and anti-DNA antibodies [26] from a systemic lupus erythematosus (SLE) individual’s serum. Antibody fragments have also been selected with the aim of viral infection inhibition, as described by Hodits et al. (1995), in which a specific scFv for low-density-lipoprotein receptor family blocks infection with rhinovirus [27]. Other examples include libraries from asthma donors [28] or individuals infected with *Streptococcus oralis* [29], hepatitis C virus [30] and Ebola virus [31].

In addition to human and mice, libraries from camelids (camels and llamas) have also become of great interest due to the size and stability of the single variable domain found in some of their natural immunoglobulins; as early as 1996 [32], camelid single-domain antibody fragments (VHH) have been considered as an alternative to new, small, and high-affinity antigen-binding molecules. Some cartilaginous fish (sharks, skates, and rays) are also considered another source of single-domain antibody fragments, as they possess an immunoglobulin-like antigen receptor, IgNAR (immunoglobulin isotype novel antigen receptor) [33]. VHH and IgNAR are used to reduce the size of functional antigen-specific molecules. These single-domain molecules have overcome the need for an interface counterpart domain, maintaining solubility, specificity, and affinity. Conventional antibodies have an interface between VH and VL and, even though these paratopes can have a very high affinity, they depend on another to prevent hydrophobic residue exposure and low solubility and expression levels [34]. Despite the diversity of species and formats of antibodies selected using Phage Display, most of the approved antibodies, including those from Phage Display, are in full-length IgG format and are of murine and human origin (Figure 1B) (Table 1). This can be explained by the ultimate use of the antibody in treating human diseases, where the effector function of the immune system is required.

Once a library has been constructed, its diversity enables the selection of antibodies virtually against any relevant target. Therefore, many libraries are built only once and used for different applications, such as the Griffiths, A.D. (1994) and Tomlison I+J synthetic scFv libraries [35], or even the one that we constructed from osteosarcoma patients [36], which was used to isolate antibodies against anti-tumoral antigens [37], α-dystroglycan mucin glycopeptide [38] and anti-Zika virus [39]. Three Phage Display commercial platforms (MorphoSys HuCal, Cambridge Antibody Technology, and Dyax) stand out as possible sources of molecules that have gone through clinical trials. It is important to note that these examples are only for monovalent antibody drugs. Many other platforms and antibody-based molecules such as CAR-T cells, antibody–drug conjugates, antibody fragments (nanobody, scFv, Fab), and bi- and tri-specific molecules have also been developed using Phage Display, as listed in the Therapeutic Antibody Database (TABS—https://tabs.craic.com/, (accessed on 05/01/23)).

## 3. Biopanning Selection for Retrieval of Specific Antibodies

Biopanning involves selection rounds to determine whether a displayed antibody can effectively interact with a bait antigen. To eliminate non-specific antibody phages, stringent methods are applied during the panning cycles [40]. Subsequently, the specific antibodies are eluted using a condition that prevents the antibody–antigen binding. The eluted phages are used to (re)infect a bacterial host, amplifying the selected phages and then initiating a new round of selection [41]. Traditionally, three to six selection rounds are performed, increasing the stringency with each selection/amplification round, resulting in an enrichment process in a Darwinian model. Over the cycles, higher-affinity antibodies become more frequent, and at the end, high-affinity antibodies to the antigen are markedly enriched [42].

The variety of ways to carry out the selection starts with a wide range of possibilities for how to present the antigen to the phage library; it can be isolated, combined with other molecules, or even arranged in a cellular structure. There are also several ways to eliminate non-binding antibodies [43]. Compared to other methods for generating monoclonal antibodies, such as transgenic mice, Phage Display technology provides researchers with great flexibility in selecting platforms, from simple to complex systems (Table 2). Among the current technologies for the development of human antibodies, Phage Display is the most suitable for obtaining antibodies against toxic antigens [14,44].

Several selection strategies have already been employed for different purposes, in the search for therapeutic antibodies, for diagnosis, industrial use, etc. (Table 2). Three selection strategies stand out: selection using immobilized molecular antigens with a huge variety of immobilization forms; selection using intact cells or biological tissues; and a more challenging strategy, the in vivo selection [76,77,78].

A commonly used method for selecting antigens involves adsorbing the antigen directly onto a surface or adding a biotinylated antigen to a surface covered with avidin, streptavidin, or a similar molecule [79,80]. This immobilizes the antigen, which is then incubated with the antibody library. This selection method is advantageous because it is simple and involves a common and inexpensive experimental methodology. Restricting the freedom degree of the (biotinylated) antigen can be useful to limit the availability of epitopes or to avoid conformations that may interfere with antibody binding in more natural situations. Synthetic peptides and complex molecules, such as viral particles, have also been used in this type of selection [65,81,82,83,84].

From this basic form of selection, other variants have been proposed, such as the capture of the antigen by a biotinylated antibody that was previously adhered to a plate with streptavidin [39]. Another variation is the selection with immobilized antigens in solution, where the biotinylated antigen is incubated, before or after binding to the antibodies, with magnetic beads coated with streptavidin or anti-biotin antibody, or with streptavidin–agarose. In the case of magnetic beads, specific antibodies are obtained by passing them through a magnetic separator (such as a magnetic column) [85,86]. In the case of streptavidin–agarose, the system undergoes a pull-down followed by centrifugation and the pellet is washed several times to completely remove the non-binding phages [59]. The selection with magnetic beads or pull-down resin is noteworthy, as it provides an additional stringent force to the system, reducing the possibility of non-specific phage remaining, and because it allows a selection in solution that provides a greater amount of antigen accessible to antibodies [76,87].

Another selection method involves the use of a compact solid Marketatrix column with the antigen immobilized, similar to liquid chromatography. This technique is called Chromato-panning and involves adding the library to the column. Non-specific phages are removed with washing, while specific phages are eluted by adding an acid or salt-saturated buffer. These specific phages can then be used for a subsequent selection round [88].

A major revolution in selection methods has been the implementation of negative selection, which consists of incubating a library of antibodies to one or more non-relevant antigens, especially antigens that are related to the antigen of interest, to remove non-specific binders [89,90]. This technique helps to reduce the possibility of contamination with non-relevant antibodies, especially in the case of structurally similar antigens. Negative selection does not necessarily have to be performed before the positive one; it can be carried out in parallel [91,92].

To select specific antibodies against a given antigen in its biological context, several studies have performed Phage Display selection using whole cells, isolated or in tissue conformation. This type of selection is often adopted against membrane antigens, whose native conformation is crucial to success in obtaining antibodies that bind to the target in vivo [68,69,93].

The cells can be presented adhered to a surface or in solution (Figure 2), and there is a wide range of methods for conducting the selection. Cells can be attached to a surface, forming a monolayer that will later receive the phage library [66,94,95], or it is still possible to use a whole biological tissue, which preserves cellular interactions and the natural presentation of antigens on the cell surface, which can be altered in the cell isolation process [96]. Dorfmueller et al. [63] isolated antibodies against corneal endothelial surface antigens. Excised and preserved corneas were incubated with the phage library and subjected to several washes to remove non-binding phages. Then, the tissue was scraped, and the specific phages were eluted using trypsin treatment [63].

A variant of whole-cell selection is BRASIL (Biopanning and Rapid Analysis of Selective Interactive Ligands) technology, which consists of performing a previous step of negative selection using the cells of interest lacking the antigen (Figure 2). The phage library is incubated with antigen-negative cells, and then, this mixture is placed in a tube filled with an organic immiscible solution that acts as a filter. This tube is then centrifuged to separate the cells carrying the binding phage (pellet) from the non-binding phages that remain in the upper aqueous solution. The non-binding phages are collected and incubated with the same cell type that expresses the antigen of interest. The mixture is added to a new tube containing the same organic solution as before and centrifuged, and the pellet is used to amplify the binding phages for a new round. This method is interesting because it allows for selecting antibodies specific to a cell surface antigen, avoiding contamination with antibodies that bind to other cell antigens [37,76,91,95].

Sorensen et al. [60] used a creative method to select antibodies specific to a cell type, performing positive and negative selection simultaneously. A hematopoietic cell (K562, of female origin) was positioned via micromanipulation in a specific region on a glass plate, close to several lymphocytes of male origin. The phage library was added to the glass plate to allow binding to the cells. A gold disc was placed on the plate using micromanipulation at the K562-specific position, and shortly after this, the plate was irradiated with ultraviolet light to inactivate the phages so that the phage binding was protected from radiation by the gold disc. The authors’ idea was to implement a micro-selection methodology that could be used to obtain antibodies specific to a rare cell type in a heterogeneous population. This work exemplifies the diversity of selection methods offered by Phage Display technology, depending on the purpose of the selection [60].

Another example of selection recently used is in vivo selection, which goes beyond whole-cell selection, and it searches for antibodies that bind to a biological antigen in its natural context within the organism (Figure 2). Despite being interesting due to its approximation to the physiological condition, this methodology greatly increases the complexity of the panning process and has posed major challenges to researchers, such as the recovery of binding antibodies and the avoidance of selection of non-specific antibodies, given the availability of a huge diversity of antigens [97,98,99].

In vivo selection has been extensively studied with peptide libraries, but there have also been some important studies with antibody libraries [40]. Hemadou et. al. [70] selected recombinant antibodies specific to atherosclerotic proteins using three selection rounds, in which an scFv-phage library was applied to rabbit models of atherosclerosis. After 1 h of phage circulation in the bloodstream, the animals were sacrificed, the cardiac route was washed with PBS to ensure the removal of non-binding phage, and the binding phages were eluted from aorta sections. Thousands of selected antibody clones were then tested using flow cytometry, and the positive clones were sequenced. The methodology adopted by the authors allowed an adequate selection for that purpose by coupling the in vivo selection to a high-throughput screening, which accelerated the panning process [70]. High-performance sequencing, combined with a powerful screening method, such as flow cytometry, improves the discovery of specific antibodies when there is a large antigen diversity [71,99].

## 4. Elution Methods for Improving the Specificity of the Antibody Selection

The elution step is crucial to select the best clones later in biopanning. Different researchers have mainly chosen one of the following three types of elution. The choice is also influenced by the library type and how the antigen is immobilized.

The first elution employed is one in which a change in pH is used once it impairs the antibody/antigen interaction, releasing the antibody. Either an acidic (glycine, citric acid) or alkaline (triethylamine) elution buffer can be used [100]. However, neutralization is required after acidic elution to avoid degradation of the phages and to preserve their infectivity [101]. pH-based elution has been the most commonly used method in panning strategies. Recently, a citrate buffer (pH 3) was used to elute phages displaying Fab fragments. The affinity of these antibodies was estimated to be in the order of 2 μM [102].

Some libraries are constructed to contain an internal cleavage site between the antibody and the phage pIII protein, enabling the release of phages displaying selected antibodies after proteolytic cleavage. Depending on the site used, proteases such as trypsin, chymotrypsin, thrombin, or factor Xa can be used [103,104]. Lysozyme has also been used as an alternative to elution via proteolytic cleavage [105]. If enzymatic digestion is used, phages that result from this step are non-infectious. Hence, there is a selection improvement of antibody-displaying phages that are still infective [106].

A more specific type of elution is competition-based. In this procedure, an excess of free, non-immobilized antigen is added to the reaction to unbind phage captured by the same antigen immobilized in any way. This elution is performed to obtain antibody-displaying phages with specificity to the antigen and to avoid contamination by non-specific antibodies [104]. Kabir et al. [107] reported high-affinity antibodies obtained via a competitive binding strategy against immobilized HM-I killer toxin, an antigenic peptide from fungal pathogens [107].

## 5. Phage Display to Select Antibodies against Cancer

Phage Display technology has been used to develop antibodies specific to tumor cells antigens, enabling the identification of tumor-specific molecular markers for diagnosis and the development of targeted therapies as alternatives to radiotherapy and chemotherapy. Monoclonal antibodies (mAbs) targeting tumor-associated antigens can combat cancer through various mechanisms, including the neutralization of oncoproteins or other soluble factors, and the destruction of tumor cells by antibody-dependent cellular phagocytosis (ADCP), antibody-dependent cellular cytotoxicity (ADCC), and complement-dependent cytotoxicity (CDC) [108]. In addition to their inherent effector mechanisms, the specificity of mAbs, along with the functional modularity of their variable and constant domains, has been cleverly exploited to guide cytotoxic molecules or engineered effector cells, enabling selective targeting of tumors.

Although one of the basic hallmarks of the malignant transformation process is genomic instability, which can lead to the generation of altered proteins [109], these alterations do not always result in neoantigens being present at sufficient density on the cell surface or uniformly across all tumor cell subgroups [110]. These features are often associated with tumors with a high capacity to evade immune control mechanisms and relapse after immunotherapeutic approaches [111,112]. Moreover, the tumor microenvironment can induce immunosuppression and create physical barriers that impede the penetration of therapeutic effector cells and macromolecules into solid tumors. The combination of tumor-induced immunosuppression and the internalization of tumor-associated antigens adds further complexity to developing effective anti-tumor antibodies, presenting a more significant challenge than initially anticipated.

### 5.1. Different Methods to Search for Anti-Tumor Antibodies

The search for oncological molecular targets, now widely facilitated by advanced tools such as mass sequencing and single-cell analysis, can be approached using the Phage Display technique in its various application formats [113]. Phage Display technology addresses some of the challenges mentioned above by enabling the exploration of complex surfaces, including whole cells, to identify protein binders associated with specific tumors or to map antibody epitopes extracted from individuals sensitized to a particular antigen [114]. Indeed, Phage Display allows the screening of antibody fragment libraries against tumor cells without prior knowledge of the differential components present in tumor and healthy cells [73]. This review highlights interesting examples of this approach in the selection methods section. One such example is the biopanning method called BRASIL, which was developed by Giordano et al. [115] and later applied by Dantas-Barbosa et al. [37] for the screening of antibody fragments [37,115]. Another example is the gold disc microselection methodology, developed by Sorensen et al. [60].

In another selection approach using whole cells and elution via immunoprecipitation, Liu et al. [116] isolated an scFv-antibody that binds to PKM2, a protein overexpressed in lung cancer cells under hypoxia. When this antibody binds to the cell, it is specifically internalized. In this case, this antibody can be used as a biomarker for hypoxia in liver cancer tissues or as a targeted payload delivery vector [116].

Another advantage of using whole cells for biopanning is the possibility to obtain different clones for several epitopes of the same target. Santora et al. [117] have developed a Fab phage library using a carcinoma cell line BT-20 to generate a cancer-specific polyclonal library capable of inducing strong ADCC and CDC after removing non-specific sequences with negative selection. This library was obtained by immunizing mice with BT-20 and selection was carried out by fixing cells on paraformaldehyde [117]. Such a strategy could be relevant against resistant malignant cells, in which downregulation of some antibody targets may play a role in becoming treatment insensitive.

Alterations in tumor cells are often associated with the overexpression of proteins that are also present in normal cells of the same or even other tissues. This leads to problems in eliciting a potent immune response against self-antigens [118] and may pose safety challenges for therapies targeting these antigens [119]. However, if the expression of these antigens is limited to a specific cell lineage that can be eliminated, it offers a valuable therapeutic opportunity. For example, for the treatment of hematological tumors like lymphoma and lymphocytic leukemia, markers like CD20, CD19, CD22, and BCMA have been used as targets for monoclonal antibodies selected with naïve and immune Phage Display libraries [108]. These antigens are expressed by B-lymphocytes and their precursors, making them intriguing therapeutic targets due to their frequent overexpression in B cell malignancies. Additionally, antibodies that bind to these antigens are not significantly internalized, further enhancing their potential for therapeutic targeting. This treatment approach aims to deplete these cells from the blood circulation to control leukemias, lymphomas, and myelomas [120,121].

Another potent intervention is the use of bispecific antibodies with a CD20 or CD19 binding portion and a second portion specific to T lymphocytes (CD3) to promote the attachment of T lymphocytes to target cells, such as tumor B cells, increasing cytotoxicity and the activation of anti-tumor response [121]. The anti-CD3 bispecific antibody, denominated as bispecific T cell engager (BiTE), has been also used for other targets, including solid tumors, containing specificity against CD3 and GCC, an antigen overexpressed in gastrointestinal cancers [122].

A widely studied anti-tumor approach uses monoclonal antibodies against growth factor receptors such as HER-2 (human epidermal growth factor receptor-2) and VEGFR (vascular endothelial growth factor receptor). Overexpression of these receptors is linked to tumor metastasis and poor prognosis in many types of cancers, such as breast, colorectal, lung, neuroblastoma, gastric and cervical cancer. Several therapeutic antibodies have been approved against this group of molecules, like trastuzumab and tamucirumab. Phage Display has been employed to select novel anti-HER2 antibodies in different formats such as single-domain antibodies [123], and to select antibodies targeting new epitopes of HER2 antibodies with diverse functional activities. These antibodies have been combined with existing anti-HER2 used in clinical trials to overcome tumor resistance observed in some patients during monotherapy [124]. The effectiveness of these antibodies depends on the binding epitope since these receptors have regions with different functions, and this property is important to consider when selecting antibodies. Shahangian and colleagues (2015) performed competitive Phage Display panning to select antibodies that efficiently block VEGF/VEGFR binding. The screening process for specific antibodies involved evaluating antibodies that showed a significant decrease in VEGF binding during a competitive ELISA in the presence of VEGFR [125].

Some therapeutic approaches with anti-tumor antibodies involve mechanisms of intracellular action. Specific binding to cytoplasmic and nuclear molecules that regulate cancer development, improvised delivery of toxins with intracellular antibodies, blocking secretion of tumor receptors on the endoplasmic reticulum, and antibody-mediated alterations in cellular signaling pathways are new forms of immunotherapy in addition to interaction with the cell membrane. Furthermore, intracellular antibodies are an alternative for diagnosis, targeting internal markers. The action of intracellular antibodies faces the difficulty of these molecules crossing the plasmatic membrane and escaping from the pathways of protein degradation. Different strategies have been explored to overcome this challenge. Phage Display is an important tool to allow obtaining smaller antibody formats, such as scFv and VHH, which are more favorable structures for transit through membranes and tissue barriers and for intracellular expression [126,127].

D’agostino et al. [128] selected, with a llama immune library, the D11 VHH antibody, specific to the Twist1 factor that inhibits the p53 protein. The activity of D11 VHH was tested in fibrosarcoma cells via intracellular expression of the antibody using lentiviral vectors [128]. Pastushok et al. [129] developed, using an immunized Phage Display library, an anti-RAD51 scFv-Fc. RAD51 is a nuclear protein involved in DNA stability and is overexpressed in many types of cancer. For the intracellular action of inhibiting RAD51, the authors fused the recombinant antibody to a cell penetrating peptide (CPP). CPPs are peptides used for membrane transit of different macromolecules. This type of association of antibodies with CPPs is called TransMab [129]. Another immunotherapy alternative in cancer is the construction of bispecific antibodies containing the anti-tumor antibody fused to a fragment of an autoantibody. Autoantibodies are known to have intrinsic capacities to cross cell membranes [130].

Phage Display in vivo selection can be an interesting alternative to select functional anti-tumor antibodies against unknown antigens that are important for tumor progression. Using a VHH library in mice with cancer, it was possible to select antibodies binding to dynactin-1-p150, which is present in tumor blood vessels of metastatic glioblastoma and is associated with the angiogenesis process, but not in normal vessels [131]. This is an example of how novel targetable proteins can be discovered to develop new anti-angiogenic antibodies that can be used in combination with ones already approved.

### 5.2. The Use of Specific Antibodies for Immune Checkpoints

Immune checkpoint pathways are relevant for tumor progression and survival and many immune-inhibitory molecules are expressed on the surface of tumor cells. These molecules, such as PD-L1, LAG-3, CTLA-4, and CD272, promote the reduction in the immune response and tumor tolerance [132,133]. PD-L1, which is primarily expressed in various immune cells such as dendritic cells and macrophages, interacts with PD-1 found on activated T cells. This interaction subsequently downregulates T cell receptor activation and serves as an immunoregulatory mechanism. However, certain tumor cells overexpress PD-L1 to inhibit T cell activity, particularly in the presence of cytotoxic T-lymphocytes. This leads to T cell exhaustion, preventing the elimination of tumor cells. Thus, the PD-1/PD-L1 interaction in the tumor infiltrate negatively regulates anti-tumor immune activity [134].

Immunotherapy with anti-PD-1 and anti-PD-L1 antibodies has been a highly successful intervention for several types of cancer, including melanoma, renal, cervical, gastric, breast and lung cancer, and it involves preventing the interaction between PD-1 and PD-L1 using antibody binding, blocking T cell suppression in solid tumors. These antibodies can also lead to the elimination of tumor cells by activating NK cells. This immunomodulation has opened up new perspectives for cancer treatment in the clinic, with some antibodies already approved, such as durvalumab (anti-PD-L1), avelumab (anti-PD-L1), and camrelizumab (anti-PD-1) [135]. Experiments have shown that anti-PD-L1 antibodies are functionally more effective in blocking T cell inhibitory signaling than anti-PD-1 antibodies [136]. Phage Display is an important tool to provide potent antibodies against the immune checkpoint molecules. Several anti-PD-1 and anti-PD-L1 antibodies were developed using Phage Display in different immunoglobulin structures, such as IgG1 and IgG4 [137,138,139]. The anti-PD-L1 antibodies, atezolizumab and avelumab, approved for urothelial carcinoma, were developed using naïve Phage Display libraries. Antibodies specific for other immune checkpoint molecules have also been studied, such as CTLA-4 [140] and LAG-3 [141].

In addition to whole antibodies, anti-PD-L1 VHHs have been isolated from naïve or immune llama Phage Display libraries [132,142]. The search for anti-PD-1/PD-L1 VHH is supported by the better tissue penetration of this small molecule, which becomes interesting for solid tumors where the diffusion of larger antibodies is limited. However, when administered in humans, VHHs derived from camelids may pose challenges that require further development to mitigate immunogenicity. To circumvent this problem, human single-domain antibodies have been developed using methods that facilitate the generation of specific variable domains with human sequences. For instance, synthetic Phage Display libraries or transgenic mice have been employed to obtain these human-specific variable single domains [143].

Another promising approach is multi-immunotherapy based on fusion proteins composed of domains derived from several proteins with diverse immune activities. In this direction, a fusion protein containing an anti-PD-L1 VHH (obtained using Phage Display), an anti-CD16a VHH, and IL-15 showed higher anti-tumor activity in vitro and in vivo comparable to the separate counterparts. The idea of this fusion protein is to block the PD-L1/PD-1 pathway, thereby maintaining T cell activation, and to attract and activate NK cells (CD16, IL-15) to induce cytotoxic activity [144].

Phage Display’s primary function in identifying specific targets and ligands goes beyond aiding the development of various oncological immunotherapies. It also propels the progress of diagnostic techniques and methods for monitoring patient responses throughout and post-treatment. Staroverov et al. [145] developed Heat Shock Protein (HSP)-specific antibodies using a human scFv phage library. Levels of HSPs in affected animals’ blood sera were evaluated through immunoassays with recombinant Phage Display-derived antibodies [145]. HSPs are produced in response to stressors such as heat and toxins. They can be overexpressed in tumor cells due to increased stress during rapid growth, leading to their release into circulation. Researchers explore their potential as cancer biomarkers for diagnosis, prognosis, and treatment response monitoring [146]. Exosomes, small extracellular vesicles that facilitate cell communication, are also investigated for diagnostic methods. They are released by various cell types, including cancer cells, containing proteins, lipids, and nucleic acids. Their presence in bodily fluids makes them candidates for liquid biopsies—a less invasive approach compared to tissue biopsies. A recent proof-of-principle study employed in vivo peptide Phage Display screening to identify markers in exosomes linked to a multiple myeloma pre-clinical model [147].

### 5.3. Expanding the Application of Antibodies: Antibody Drug Conjugates (ADCs) and Chimeric Antigen Receptors (CARs)

Two notable advancements in the field of mAb-based oncological treatment have gained significant attention in recent years, fostering high expectations for the future. One of these advancements involves the conjugation of antibodies to cytotoxic payloads, such as highly potent drugs or radioisotopes. This enables the specific addressing of these active molecules to act on the intended target [148,149]. Another advancement involves therapies utilizing engineered cells that express a chimeric antigen receptor (CAR) on their surface. This receptor consists of portions derived from various molecules, including an extracellular portion derived from the variable regions of mAbs [150]. The chimeric receptor enables the recognition of a specific antigen presented on the tumor cell surface, directing CAR-harboring effector cells to the target [151].

In these approaches, similar to naked mAbs, the pursuit of efficient and specific antibodies targeting unique or differentially expressed tumor antigens remains paramount. However, in these cases, there is a need for more stringent considerations regarding the acceptable threshold for on-target off-tumor effects. The cytotoxic potential of these systems, involving highly potent and non-specific payloads in the case of conjugates, is intentionally designed to be more aggressive than that of naked mAbs [152]. In the context of the therapy with CAR-expressing cells, the heightened cytotoxicity of activated effector cells could lead to significant and long-term side effects associated with off-tumor toxicity, as they are induced to proliferate, persist, and potentially generate memory cells [153,154,155].

#### 5.3.1. Examples of Antibody Drug Conjugates (ADCs)

The efficacy and safety of antibody conjugates are strongly influenced by the conjugation strategy employed, which includes the choice of chemical groups involved in the covalent linkage between the payloads and the antibody. The conjugation strategy determines the payload-to-antibody stoichiometry and the degree of homogeneity within the conjugate pool produced in each batch [156]. Depending on the number of loaded molecules and their conjugation sites on the antibody, the system may exhibit variable pharmacokinetic profiles, potentially impacting its stability and even affecting the kinetic parameters of the antibody interaction with the antigen [157]. While all antibody drug conjugates (ADCs) approved by the FDA to date use random conjugation methods based either on thiol coupling to reduced cysteines or amine coupling to lysine residues exposed on the the surface of the antibody, efforts are ongoing to advance new generations of conjugates by focusing on site-specific conjugation strategies [158]. A commonly employed method is the substitution of specific endogenous residues with cysteines, leveraging the reactivity of their sulfhydryl groups while preserving the interchain disulfide bonds of the antibody [159]. This approach allows for targeted modification or conjugation of the antibody while maintaining its structural integrity. However, the criteria for determining an ideal residue for substitution with a cysteine prone to conjugation are not fully established.

An interesting example of the use of Phage Display to identify protein residues suitable for substitution with cysteines prone to conjugation was presented by Junutula and co-workers at Genentech (2008). In their work, which introduces the method called PHESELECTOR, a library of variants with different points of cysteine substitution in the Fab fragment of trastuzumab (anti-HER2) displayed on phage was subjected to conjugation with a thiol-reactive biotin payload. The library was subsequently selected using ELISA on plates coated with streptavidin and HER2, in separate steps, to identify phages that were successfully conjugated to the payload without losing the ability to bind to the antigen [160]. Based on insights into the optimal sites for the inclusion of conjugation-prone cysteines, highly pure ADCs with a drug-to-antibody ratio of 2.0 were successfully generated via site-directed mutagenesis. Genentech’s THIOMABs (the full mAbs generated using this method) exhibited higher tolerance at significantly increased doses in animals and a slower clearance rate from circulation in rats, compared to conventional ADCs [161]. Moreover, the cysteine–maleimide linkage of THIOMAB displayed superior serum stability [162]. These studies demonstrated the feasibility and advantages of site-specific antibody–drug conjugation, prompting a surge of activity in the development of site-specific ADCs.

The FDA-approved anti-CD22 Lumoxiti (moxetumumab pasudotox-tdfk) (Table 1) is classified as an ADC. It comprises an engineered murine scFv genetically fused to a modified Pseudomonas toxin. Using Phage Display, the CDRH3 of the original mAb was enhanced in its affinity for CD22, resulting in a 14-fold increase [163]. CD22 is expressed on the surface of various malignant B cells, including hairy cell leukemia (HCL), the current approved use of Lumoxiti [164].

In the context of lung cancer (solid tumor), a conjugate consisting of a C4.4A-targeting human antibody (lupartumab) developed via Phage Display from a human synthetic combinatorial library and a highly potent microtubule-disrupting drug (amadotin) has been evaluated (NCT02134197) [165]. C4.4A (LYPD3) is overexpressed in non-small-cell lung cancer (NSCLC) and is associated with a malignant phenotype and poor prognosis in NSCLC [166]. Under normal conditions, C4.4A is primarily expressed in stratified squamous epithelia. However, knock-out mice lacking C4.4A show no significant abnormalities, and their epidermal development is normal [167]. Developing a C4.4A-targeting ADC represents a promising opportunity for the selective treatment of C4.4A-positive tumors.

#### 5.3.2. Chimeric-Antigen-Receptor-Expressing Cells: Design and Targeting Strategies

CARs are synthetic molecules created by the genetic combination of a variety of functional domains from different immune receptors. These engineered genes are then inserted into effector cells, typically autologous T lymphocytes, which are modified outside the body and reinfused into the patient to selectively redirect these effector cells to recognize and attack tumor cells [168]. In addition to CAR-T cell therapy, which already includes six FDA-approved products, all for hematological cancers, other effector cells such as natural killer (NK) cells [169] and macrophages [170] have also been engineered with CARs and tested in pre-clinical and clinical trials [171], expanding the potential applications of this therapy. Regarding their structure, several combinations of intracellular, transmembrane, and extracellular domains from different sources have been extensively explored to construct improved CARs. This design and redesign process is crucial as each component can potentially affect the performance of the therapy, either independently or synergically [172]. Crucial steps include the appropriate selection of the extracellular domain responsible for specificity, typically an scFv, as well as the intracellular domains responsible for intracellular signaling that triggers activation upon antigen encounter on the target cell surface, regardless of its presentation via MHC. Similar to the natural activation of T cells, CAR-T cells also rely on the integration of multiple stimulatory signals [173]. This process is initiated through the CD3 protein, an essential accessory component of the TCR complex. The CD3ζ chain acts as the main intracellular signaling domain in these receptors. Furthermore, basic costimulatory proteins are required for proper cell activation in the natural context and have been incorporated into CARs through improved generations of these molecules [174], which will be discussed later in this section.

Despite the high efficacy demonstrated in hematological tumors, therapy with CAR-expressing cells still faces several challenges in its application to treat solid tumors [175]. One challenge is the infeasibility of completely depleting cells expressing the target antigen, as solid-tumor-associated antigens are often found in healthy cells of vital tissues. However, several proteins overexpressed in tumor cells have been tested as targets for CAR cell therapy [175], including mesothelin (MSLN), a membrane glycoprotein expressed in normal mesothelial tissues but highly expressed in various malignancies, such as non-small-cell lung cancer (NSCLC) and ovarian cancer. MSLN has been investigated as a target for CAR-T cells in treating solid tumors, including mesothelioma, lung, breast, and pancreatic cancer [175,176]. Recently, Chen et al. [177] selected a human anti-MSLN scFv using Phage Display, which was subsequently used to construct a second-generation CAR. The phage-derived anti-MSLN CAR-T cells demonstrated in vitro efficacy against ovarian tumor cell lines and in vivo effectiveness against ovarian cancer cell-derived xenografts. Preliminary clinical trials using the novel anti-MSLN CAR-T cell therapy in autologous T cells from three patients with chemotherapy-refractory metastatic ovarian cancer showed promising results, indicating its potential as an immunotherapeutic approach for advanced ovarian cancer [177].

Rhabdomyosarcoma (RMS) is a common soft tissue tumor in children, traditionally treated with a highly toxic chemotherapy regimen, which has shown limited efficacy in improving the survival of patients with metastatic or relapsed RMS [178]. Despite the high immunologic escape capacity observed in some forms of pediatric RMS, immune checkpoint blockade has not been proven to be an effective strategy, a characteristic shared with other low-mutation-rate pediatric tumors [179]. The immunosuppressive microenvironment, characterized by a collagen-rich stroma, has been attributed to the activity of myeloid cells in response to T cell activity in the tumor. Despite the limited availability of neoantigens in RMS, the upregulation of fibroblast growth factor receptor 4 (FGFR4) in these tumors has identified it as a potential target for new-generation immunotherapies as an alternative to highly toxic conventional chemotherapy [180]. In this context, Sullivan et al. [180] reported the construction of CARs targeting FGFR4 developed by using human Fab and VH libraries to select anti-FGFR4 fragments via Phage Display. The inefficiency of T cells engineered with CARs containing previously selected scFvs against the full-length protein led the group to improve the biopanning strategy by including selection rounds using only the FGFR4 domain closest to the cell membrane, which likely enhances the performance of CAR-T cells even against tumor cells with low target molecule density. With the new construct, the anti-FGFR4 CAR-T cells combined with a multi-drug approach targeting the myeloid cells responsible for the immunosuppressive microenvironment showed high tumor elimination capacity in orthotopic mouse models [180].

Another challenge in CAR-T cell immunotherapy is the use of these engineered cells against T cell tumors, as both cell types share a similar molecular phenotype. An alternative is the use of intracellular antibodies, which can be selected using Phage Display, in T cells in addition to the CAR gene modification, to block the expression of target molecules in the effector cell. Thus, CAR-T cells can act against tumor T cells by specifically binding the chimeric receptor, without the effects of self-damage [181].

The impact of the antibody-derived segments of CARs, including factors such as immunogenicity and affinity, is still under ongoing research. Studies suggest that scFvs with moderate affinity can enhance CAR-T cell performance in certain contexts. Furthermore, immunogenicity resulting from the murine origin of scFvs in approved CARs has also been investigated [182,183]. Phage Display offers means to search for scFvs with advanced properties in all these contexts, as well as to identify new targets and epitopes.

Many advanced improvements have been made in CAR signaling domains, leading to the emergence of different CAR generations. First-generation CARs, which contain an intracellular CD3ζ chain, activate T cells but exhibit limited proliferative activity and cytotoxicity. Second-generation CARs, incorporating an additional costimulatory molecule (CD28, 4-1BB, or OX40), exhibit enhanced proliferation, cytokine release, durability, and stability [172,173,174]. Further generations involve the combined expression of other effector molecules, such as cytokines and mAbs, by CAR-expressing cells [184]. For instance, CAR-T cells against carbonic anhydrase (an enzyme that is overexpressed in many hypoxic solid tumors) were engineered to secrete anti-PD-L1 antibodies, and these modified CAR-T cells showed lower expression of exhaustion molecules (PD-1, TIM-3, and LAG3) after contact with tumor cells, compared to the conventional anti-carbonic anhydrase CAR-T cells. The anti-PD-L1 antibody-secreting CAR-T cells also conferred greater protection against renal carcinoma in murine models, and the secretion of the antibody led to greater recruitment of NK cells to the tumor [134].

## 6. Conclusions

Recent advances in cancer therapy have been boosted by the application of antibodies derived from Phage Display libraries. The importance of this approach has been recognized by the scientific community through the 2018 Nobel Prize in Chemistry “for the Phage Display of peptides and antibodies”, awarded to George P. Smith and Gregory P. Winter. This methodology involves biopanning of specific antibodies that can bind to different targets, such as cancer cell markers, with remarkable specificity and affinity. The potential to screen extensive libraries of phage-displayed antibodies, along with numerous available selection approaches, underscores the increasing promise of utilizing phage-derived antibodies as a highly favorable resource for advancing therapeutic product development focused on cancer-specific antibody treatments. Despite being almost 40 years old and amidst the ongoing methodological progress in target and ligand discovery, Phage Display technology remains a crucial tool in various cutting-edge research fronts, being part of the horizon of perspectives and innovations in oncology.

## Figures and Tables

**Figure 1 viruses-15-01903-f001:**
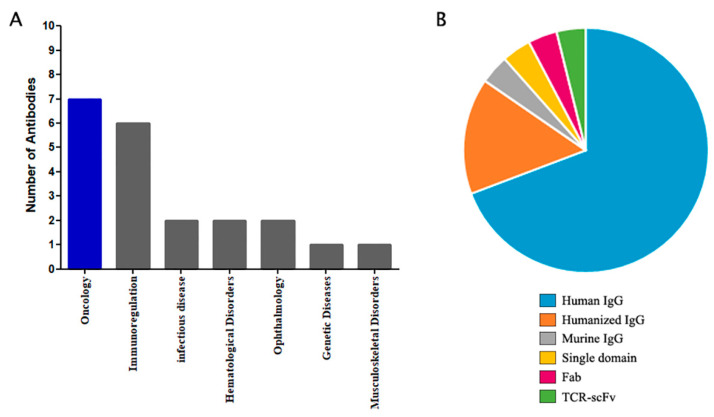
Statistics of antibodies derived from Phage Display that are approved for clinical use. (**A**) Number of approved antibodies per clinical area. (**B**) Proportion of each type of antibody.

**Figure 2 viruses-15-01903-f002:**
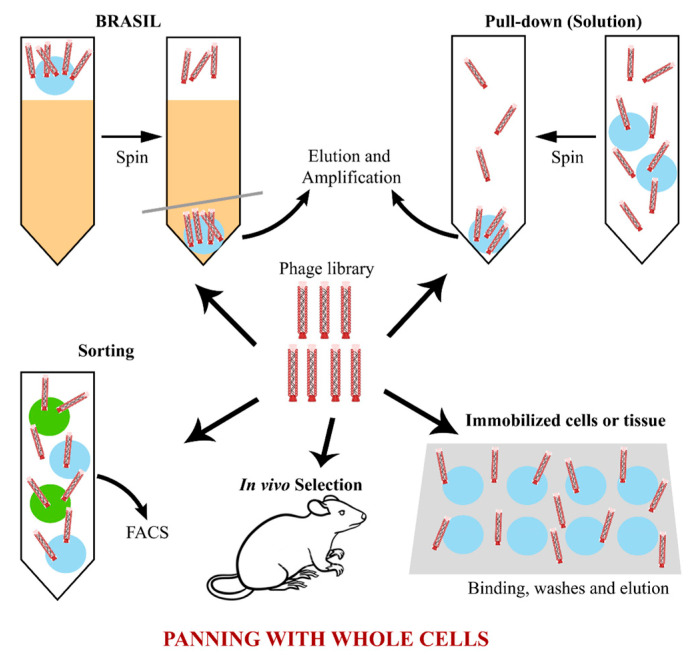
Summary of the main phage selection strategies using whole cells. It is possible to perform the panning with isolated cells, adherent or in solution, with biological tissues or with whole organisms. Each of these forms has its own advantages and difficulties, and they may provide additional selection forces using a simple pull-down method, gradient centrifugation (BRASIL) or other techniques, such as FACS or mass spectrometry.

**Table 1 viruses-15-01903-t001:** Phage Display-derived antibodies that have been approved for clinical use.

Name	Commercial Name	Format	Target	Company	Clinical Indication	Status	Phage Display Type
Adalimumab	HUMIRA	Human IgG1	TNFα	Abbvie	Crohn’s disease, Rheumatoid arthritis, Colitis ulcerative, Psoriatic arthritis, others.	Market	Naive human Fab library
Anti-MIF (BaxF07)		Human IgG4	Myostatin	Baxter	Muscle loss	Withdrawn	Naive human Fab Library
Atezolizumab *	Tecentriq	Humanized IgG1	PD-L1	Roche, Genentech	Cancers, Melanoma, Solid Tumors, hepatocellular carcinoma, others	Market	Human Library
Avelumab *	Bavencio	Human IgG1	PD-L1	Merck, Pfizer	Renal cell carcinoma, ovarian cancer, gastric cancer,	Clinical Trials. Market: Renal carcinoma	Naive Fab library
Belimumab	Benlysta	Human IgG1	TNFSF13B	GSK, HGSI	Systemic lupus erythematosus (SLE), vasculitis	Market: SLE; Phase III: Vasculitis	Human naive scFv library
Brodalumab	Kyntheum, Siliq	Human IgG2	IL-17RA	Valeant Pharmaceuticals; LEO Pharma	Plaque psoriasis	Market	MorphoSyś HuCAL
Ecallantide	Kalbitor	Not Antibody	Plasma kallikrein	Dyax Corp.	Hereditary angioedema	Market	Dyax platform
Emapalumab	Gamifant	Human IgG1	IFNγ	NovImmune	Primary hemophagocytic lymphohistiocytosis	Market	Human scFv libraries
Faricimab	Vabysmo	Human/Humanized IgG1	VEGF-A, Ang-2	Roche	Macular degeneration	Approved	Synthetic human Fab library
Guselkumab	TREMFYA™	Human IgG1	IL23A	Morphosys (Germany); Morphosys and Janssen Biotech (USA)	Psoriatic arthritis, psoriasis, palmoplantar pustulosis, others	Approved	MorphoSyś HuCAL
Ixekizumab	Taltz	Humanized IgG4	IL17A	Eli Lilly (USA)	Plaque psoriasis, psoriatic arthritis	Market	Immune murine Fab library
Lanadelumab	Takhzyro	Human IgG1	Plasma kallikrein	Dyax Corp	Hereditary angioedema attacks	Market	Naive human Fab library
Necitumumab *	PORTRAZZA™	Human IgG1	EGF Receptor	Eli Lilly (USA)	Non-small-cell lung cancer, metastatic colorectal cancer	Approved	Naive Fab library
Moxetumomab *	Lumoxiti^TM^	Mouse Exotoxin A-IgG1	CD22	MedImmune/Astrazeneca	Hairy cell leukemia	Approved	scFv library
Caplacizumab	Caplivi^TM^	Humanized VHH	VWF	Sanofi/Ablynx	Thrombotic thrombocytopenic purpura	Approved	Immune camelid nanobody library
Inebilizumab	Uplizna	Humanized IgG1	CD19	AstraZeneca/ Medimmune, Viela Bio	Neuromyelitis optica spectrum disorder	Approved	
Ramucirumab *	CYRAMZA™	Human IgG1	VEGF Receptor 2	Eli Lilly (USA)	Breast cancer, gastric cancer, lung cancer, metastatic colorectal cancer, others	Approved	Naive Fab library; Dyax platform
Ranibizumab	Lucentis	Humanized Fab	VEGF-A	Genentech	Neovascular (wet) age-related macular degeneration	Market	Fab library
Raxibacumab	ABthrax^®^	Human IgG1	Anthrax protective antigen (PA)	GlaxoSmithKline (UK); Human Genome Sciences Inc (HGSI/USA)	Anthrax inhalation	Market	Naive human scFv library; CAT platform
Relatlimab *	Opdualag	Human IgG4	LAG-3	Bristol-Myers Squibb	Melanoma	Approved	Naive human scFv phage library
Tafasitamab *	Monjuvi	Humanized IgG1/2	CD19	Xencor	Non-Hodgkin’s lymphoma, chronic lymphocytic leukemia	Approved	MorphoSyś HuCAL
Regdanvimab	Regkirona	Human IgG1	SARS-CoV-2 (spike)	Celltrion Healthcare Hungary Kft	COVID-19	Approved	Human scFv library
Tebentafusp *	KIMMTRAK	Human TCR-scFv	gp100, CD3	Immunocore	Metastatic uveal melanoma	Approved	Synthetic TCR-phage library
Tralokinumab	Adbry™ (USA); Adtralza^®^ (EU)	Human IgG4	IL-13	LEO Pharma	Atopic dermatitis, asthma	Approved	Synthetic human scFv library

TNFα: Tumor necrosis factor alpha. IFNγ: Interferon gamma. VEGF: Vascular endothelial growth factor. EGF: Epidermal growth factor. Ang-2: Angiopoietin 2. LAG-3: Lymphocyte-activation gene 3. IL: Interleukin. PD-L1: Programmed death ligand. VWF: Von Willebrand factor. gp100: A melanoma-associated antigen. Asterisk: Phage-Display-derived antibodies approved for cancer and other clinical applications.

**Table 2 viruses-15-01903-t002:** Representation of the different selection strategies and elution methods used in antibody Phage Display.

Library	Method	Reference
Synthetic scFv library	Selection in immunotubes coated with antigens. The phage preparation was added to tubes and after extensive washing, the bound phage was eluted with triethylamine.	[45]
Combinatorial naive scFv	Selection against an immobilized antigen on a roller bench. Bound phages were eluted with triethylamine.	[46]
Synthetic Fab library	Selection with seven rounds of panning, using trypsin elution.	[47]
Combinatorial scFv library	Anti-Golgi selections were carried out on streptavidin-coated magnetic beads. Biotinylated Golgi membranes were bound to streptavidin magnetic beads. After incubation, phages were eluted twice by incubating beads with triethylamine.	[48]
Combinatorial scFv library	Selection with immobilized antigens, reducing the antigen concentration over the rounds.	[49]
Combinatorial scFv library	Selection of specific antibodies against human platelets. Phages bound to the platelets were eluted with two distinct strategies: (1) Standard elution via incubation with 0.1 M glycine (pH = 2.2), followed by neutralization with Tris HCl (pH = 8). (2) Competitive elution via incubation with highly saturating concentrations (2 mg/mL) of abciximab, an anti-platelet competitor antibody.	[50]
Immune scFv library	Selection of antibodies, using sulfatide liposomes as antigen.	[51]
Combinatorial scFv library	Selection using inactivated measles virus as antigen	[52]
Combinatorial Fab library	Selection on streptavidin magnetic beads and biotin-conjugated antigen. Separation of the bound phages via pull down of the beads. Elution via competition through use of saturated antigen solution.	[53]
Human scFv library	Panning after a first negative selection with control antigen. Unbound phages were subjected to further incubation with VZV cell lysate antigen. The bound phages were eluted with 500 mL triethylamine followed by neutralization with 500 mL Tris/HCl pH 5.5, and they were amplified for a new round of selection.	[54]
Human scFv library	Panning decreasing concentrations of the antigen. The bound phages were detached via the direct addition of 1 mL of XL1-Blue cells.	[55]
Naïve scFv library	Selection against adherent IGROV-I cells. The cells were incubated with pre-warmed medium at 37 °C for 15 min to allow endocytosis of surface-bound phage. To remove phage bound to the extracellular matrix or to the culture plate, adherent cells were trypsinized. Subsequently, to remove phage bound to the cell surface, the cells were stripped three times with low-pH glycine buffer (150 mM NaCl, 100 mM glycine pH 2.5). Internalized phages were recovered from within the cells by lysing with 100 mM triethylamine for 4 min at 4 °C and neutralizing with 0.5 M TrisHCl.	[56]
Three scFv libraries	The phage library was incubated with 10^8^ PBMC. Cells were then centrifuged, washed three times, and labeled with anti-BDCA3 antibody (dendritic cells). Approximately 10^5^ cells were purified with magnetic-activated and subsequently fluorescence-activated cell sorting. Sorted cells were washed once with PBS, and cell-bound phages were eluted with HCl solution and then neutralized.	[57]
Immune mice scFv-Fc library	Phage library was used against rat liver endothelial cell plasma membranes.	[58]
Synthetic scFv library	Antibody selection, using biotinylated antigen captured by streptavidin–agarose resin. Selection can take place through centrifugation or on a column, separating the non-specific phages and the binding phages.	[59]
Commercial scFv library	Selection using cells containing the antigen of interest and control cells that are arranged on a flat surface (slide). Cells containing the antigen were previously and partially covered with a gold disc. Phages were incubated with both cells. Subsequently, non-target-cell binding phages were inactivated with UV irradiation. During irradiation, the phages bound to the antigen-bearing target cell were protected by the gold disc (radiation shielding).	[60]
Combinatorial immune Fab library	BRASIL selection. The phage library was incubated with normal cells and this system was applied in the top of an organic matrix, which was centrifuged. After, the supernatant with unbound phages was incubated with cancer cells, in the same method. Then, the cell pellet with bound phages was used directly to infect bacterial cells.	[37]
Immune llama VHH library	Masked selection: This selection consisted of incubating the phage library with antigens containing non-relevant (unwanted) epitopes, and then the non-relevant binding phages were eluted, and their antibody fragments were produced in soluble form. A new round of selection occurred, incubating antigens containing the relevant (desired) epitope with the non-relevant soluble antibody population to block (mask) the non-relevant epitopes; then, the original phage library was added to the selection.	[61]
Commercial combinatorial Fab library	Two libraries, one with Kappa chain and the other with Lambda chain, were mixed and used to select antibodies against protein F pre-fusion and post-fusion of RSV, performing a negative selection with pre-fusion protein and a positive post-fusion, and, in parallel, selection in reverse (negative for post-fusion and positive for pre-fusion).	[62]
scFv naive library	Selection of binding phages to a monolayer of corneal epithelial cells and elution with trypsin, with the support of a peristaltic pump.	[63]
Synthetic human VHH library	Selection of specific phages using a streptavidin-based Mass Spectrometry Immunoassay (MSIA Streptavidin D.A.R.T.) selection method. Metal pins containing streptavidin were attached to biotinylated antigen. The mixture was incubated with the phage and then washed to remove non-binding phage. Each wash cycle consisted of direct injection into an LC-MS in pulses with increasing speed. Afterward, the metal pins were incubated with a glycine-HCl solution, and the eluted phages were subjected to 300 cycles of low-speed injection and dispensing. The eluted fractions were immediately neutralized.	[64]
Synthetic scFv library	Selection with astringency with increasing Tween concentration, instead of the number of washes.	[65]
Immune rabbit scFv library	Selection with whole immobilized cells of the bacteria *Campylobacter jejuni*.	[66]
Naive scFv library	Selection with brain epithelial cells in solution.	[67]
Synthetic scFv library	Affinity column selection. Live *Listeria monocytogenes* cells were immobilized on a matrix in an immobilization column. The phage library was applied in the column. Elution was conducted with acid solution and centrifugation.	[68]
Synthetic scFv library	Selection of internalized phages in breast cancer cells. The phage library was incubated with cells and after removing unbound phages via washes, the specific phages were internalized. The internalized phages were eluted using cell lysis.	[69]
Synthetic scFv library	In vivo selection of atherosclerotic aorta antigen-specific scFv. The phage library was injected into ill rabbits and the phages circulated for one hour. The animals were sacrificed, and PBS was applied to ensure removal of free phages in the blood. The aorta was removed, and the phages were eluted.	[70]
Naive scFv library	In vivo selection in mice, with 3 h of phage circulation. Phages binding to tumor organs (lung and breast) and control organs (brain, muscles, pancreas) were analyzed.	[71]
Naive scFv library	Selection against immobilized antigen with astringency by gradually increasing intense washes in each round.	[72]
Combinatorial naive scFv library	Selection of specific antibodies against a biotinylated antigen, immobilized on a streptavidin surface, using two elution strategies: (1) Standard elution via incubation with 0.1 M glycine (pH = 2.2), followed by neutralization with Tris HCl (pH = 9). (2) Competitive elution via incubation with highly saturating concentrations (2 mg/mL) of non-biotinylated antigen (free antigen).	[39]
Immune murine scFv library	Selection of antibodies against a mixture of leukocyte cells from leukemia patients, with a previous depletion step with leukocyte cells from healthy donors. In the selection, the non-specific phages were removed with centrifugation and the cell-surface-bound phages were eluted with trypsin.	[73]
Naïve Fab library	Selection using a Phage Display library, in which the Fabs are attached to the phage by a covalent bond with a capture protein (SpyCatcher) expressed on the surface of the phage, instead of the classic form of gene fusion of the antibody gene and the gene of a virus protein, avoiding the subcloning step of the library construction (smaller phagemids). The selection was performed with different times of incubation of the library,, with an immobilized antigen in each round.	[74]
Synthetic camelid nanobody library	Panning using a single selection round, with a strong astringency method of twenty washes with PBS and four additional washes with acid buffer. The bound phages were eluted via competitive elution with the free antigen.	[75]

## Data Availability

Not applicable.

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
