# Peer review of "Progress on Phage Display Technology: Tailoring Antibodies for Cancer Immunotherapy"

_viruses, 2023, doi:10.3390/v15091903_

Round 1
Reviewer 1 Report
The authors provided a well-documented overview of the impact of phage display technology in the identification of tissue-specific antibodies. In addition, this article provides an interesting starting point for a discussion on the next challenges for phage-passionated researchers. The article is well written, but It could be improved by expanding the conclusions and adding a personal point of view regarding the future potential and challenges of phage display technology: characterization of extracellular vesicles subpopulations (PMID: 35141731 and others), design of theranostic strategy targeting tyrosine kinase membrane receptors (PMID: 33918836, PMID: 37241784), phage therapy (PMID: 37604859, PMID: 37164015) are just examples of the possible authors speculations.
Good luck!
Minor editing of the English language required
Author Response
We are grateful for the reviewer's insightful improvement suggestions. Given the precise scope established for this article, which centers on antibodies developed through phage display within the context of cancer, we have made a considered decision to omit certain instances of application examples. Even so, we have included an additional paragraph at the end of Section 5, presenting two examples that highlight the substantial contribution of phage display technology to the advancement of diagnostic methodologies and the monitoring of patient responses to therapy, particularly in relation to the topic of liquid biopsy—a frontier area within oncological research (please refer to lines 503 to 519).
Lines 503-519: “Phage display’s primary function in identifying specific targets and ligands goes beyond aiding the development of various oncological immunotherapies. It also propels the progress of diagnostic techniques and methods for monitoring patient responses throughout and post-treatment. Staroverov et al. (2022) developed Heat Shock Protein (HSP)-specific antibodies using a human scFv phage library. Levels of HSPs in affected animals' blood sera were evaluated through immunoassays with recombinant phage display-derived antibodies [145]. HSPs are produced in response to stressors such as heat and toxins. They can be overexpressed in tumor cells due to increased stress during rapid growth, leading to their release into circulation. Researchers explore their potential as cancer biomarkers for diagnosis, prognosis, and treatment response monitoring [146]. Exosomes, small extracellular vesicles that facilitate cell communication, are also investigated for diagnostic methods. They are released by various cell types, including cancer cells, containing proteins, lipids, and nucleic acids. Their presence in bodily fluids makes them candidates for liquid biopsies – a less invasive approach compared to tissue biopsies. A recent proof-of-principle study employed in vivo peptide phage display screening to identify markers in exosomes linked to a multiple myeloma pre-clinical model. [147]”
In response to the core essence of the pertinent feedback, we have expanded the conclusion (lines 699-702) to underscore the enduring resilience of phage display technology in light of emerging methodologies. This expansion highlights its continued relevance and effectiveness in advancing the field within the boundaries set for this review.
Lines 699-702: “Despite being almost 40 years old and amidst the ongoing methodological progress in target and ligand discovery, phage display technology remains a crucial tool in various cutting-edge research fronts, being part of the horizon of perspectives and innovations in oncology.”
We also improved the writing, editing the English.

Reviewer 2 Report
Summary / significance: The importance of isolating therapeutic antibodies with functional properties is increasing with the discovery of biomarkers. The classical animal immunization procedure and generation of monoclonal antibodies is complicated and time-consuming. This review by Alves Franca et al. summarizes articles on various technical approaches using phage display to isolate antibodies functional in cancer therapy.
Level of interest/merit: Understanding the capabilities and limitations of phage display technology is critical to isolating novel powerful tools to use e.g. in anti-tumor therapy. This is an interesting, extensive and well-written review focussing on technical summaries and studies that aimed to derive functional antibodies in oncological contexts. The article is very extensive, the information of techniques is comprehensive, the literature covered is huge, and the level of background is high.
Comment: This review is informative and easy to read for the expert, while more challenging for the beginner in the field. As such, I have some suggestions to make that I hope will improve the paper: Several structural improvements could be addressed to make the article more clear to read, and some informations could be added in additional Figures and Tables to better visualize the contents.
Specific comments:
1) Essentially, the English phrasings needs thorough improvements. often the word order is inappropriate, redundant and complicated. Especially in the first part of the manuscript. I have uploaded an edited pdf version with a number of proposed changes. If possible, the article should be edited and proofread by an English native speaker or expert.
2) Subchapter headings and figure legends could be more informative and detailed. For example, subchapter title 4, line 310, could be more stringent. Line 338ff, chapters 5, contains the main body of the manuscript. 5.1., line 363m 5.2. line 430, 5.3.1. line 501, 5.3.2. line 555 subtitles are very short and not descriptive. The authors should come up with some more extensive contents here. The abbreviations ADC and CAR (lines 501 and 555), though explained before, could be written out when mentioned in a subheader again.
3) in lines 405ff and line 425 it is not mentioned, which tumors express HER2 and/ or VEGFR and dynactin. Also in line 544 it remains unclear, which solid tumors are meant. A summary or table listing the achieved findings of phage display antibodies against diverse cancer antigens concluded from the articles could be amended.

Essentially, the English phrasings needs thorough improvements. often the word order is inappropriate, redundant and complicated. Especially in the first part of the manuscript. I have uploaded an edited pdf version with a number of proposed changes. If possible, the article should be edited and proofread by an English native speaker or expert.
Author Response
1) Essentially, the English phrasings needs thorough improvements. often the word order is inappropriate, redundant and complicated. Especially in the first part of the manuscript. I have uploaded an edited pdf version with a number of proposed changes. If possible, the article should be edited and proofread by an English native speaker or expert.
Answer: We appreciate the carefully edited and complete version that was provided by the reviewer. We incorporated the suggested changes in the article. We also improved the writing, restructuring sentences and improving our English.
2) Subchapter headings and figure legends could be more informative and detailed. For example, subchapter title 4, line 310, could be more stringent. Line 338ff, chapters 5, contains the main body of the manuscript. 5.1., line 363m 5.2. line 430, 5.3.1. line 501, 5.3.2. line 555 subtitles are very short and not descriptive. The authors should come up with some more extensive contents here. The abbreviations ADC and CAR (lines 501 and 555), though explained before, could be written out when mentioned in a subheader again.
Answer: We are thankful for the suggestion and we rewrote the titles and subtitles with more information, lines 180, 307, 334, 358, 457, 521-522 and 545, and more details in the abbreviations, lines 545 and 598.
Line 180: “3. Biopanning selection for retrieval of specific antibodies”
Line 307: “4. Elution methods for improving the specificity of the antibody selection”
Line 334: “5. Phage display to select antibodies against cancer”
Line 358: “5.1. Different methods to search for anti-tumor antibodies”
Line 457: “5.2. The use of specific antibodies for immune checkpoints”
Lines 521-522: “5.3. Expanding the application of antibodies: Antibody drug conjugates (ADCs) and chimeric antigen receptors (CARs)”
Line 545: “5.3.1 Examples of antibody drug conjugates (ADCs)”
Line 598: “5.3.2 Chimeric antigen receptor-expressing cells: design and targeting strategies”
3) In lines 405ff and line 425 it is not mentioned, which tumors express HER2 and/ or VEGFR and dynactin. Also in line 544 it remains unclear, which solid tumors are meant. A summary or table listing the achieved findings of phage display antibodies against diverse cancer antigens concluded from the articles could be amended.
Answer: We are grateful for the relevant suggestion. We detailed the specific types of cancers that express the antigens mentioned, in lines 410-411 and 451-452. We also made clearer that lung cancer is the type of solid cancer that overexpresses the C4.4A antigen, line 587. Following the reviewer’s suggestion, we specified what kinds of cancer express the antigens CD20, CD19, CD22, and BCMA in line 392 and what cancers are involved with the immunotherapy with anti-PD1 and PD-L1 antibodies in lines 471-472.
Note that the first table contains a relation between the approved antibodies that were developed by phage display and their specific tumor antigens, as well as the type of cancer with these antigens.
Thus, the table and the information in the text describing anti-cancer antibodies generated by phage display in this review presents an overview of the different tumor antigens that are explored by immunotherapy and the types of target cancers.

Reviewer 3 Report
1. It is well composed, adequately referenced with relevant to the topic, and well written comprehensive review.
2. Lines 48-53 are redundant and can be combined to once (not repeating).
3. Replace word Different in line 59, to several or various.
4. Line 123, close the parenthesis and a period.
5. Can also comment on as human polyclonal antibody vectors (to combat escape variants as compared to a monoclonal antibody) grafted with Fab fragments that were selected from phage display (either naive or immunized with tumor cells).
Author Response
- It is well composed, adequately referenced with relevant to the topic, and well written comprehensive review.
- Lines 48-53 are redundant and can be combined to once (not repeating).
Answer: We corrected the redundant information and added a phrase in lines 52-54.
Line 52-54: The selection of the phage display involves searching for antibodies with specificity to an antigen of interest, in a vast antibody ocean that consist of the libraries (immune, non-immune or synthetic) of antibodies [5].
- Replace word Different in line 59, to several or various.
Answer: We replaced the word “Different” to “Several” in line 60.
- Line 123, close the parenthesis and a period.
Answer: We corrected the structural problems in line 126.
Line 124-126: One of these approaches relies on the use of directed evolution to select high-affinity mutants from a scFv library that has been randomly mutated through error-prone PCR (epPCR), as previously described [18,19].
- Can also comment on as human polyclonal antibody vectors (to combat escape variants as compared to a monoclonal antibody) grafted with Fab fragments that were selected from phage display (either naive or immunized with tumor cells).
Answer: We are thankful for the suggestion. We have added a paragraph, in lines 379-386, discussing the selection of clones against different epitopes on the same target, with the aim of avoiding resistance variants in the context of cancer.
Lines 379-386: Another advantage of using whole cells for biopanning is the possibility to obtain different clones for several epitopes of the same target. Santora et al (2000) have developed a Fab phage library using a carcinoma cell line BT-20 to generate a cancer-specific polyclonal library capable of inducing strong ADCC and CDC after removing non-specific sequences by negative selection. Such library was obtained by immunizing mice with BT-20 and selection was carried out by fixing cells on paraformaldehyde. Such strategy could be relevant against resistant malignant cells, in which downregulation of some antibody targets may play a role in becoming treatment insensitive.

Reviewer 4 Report
This manuscript is an excellent review written by Maranhão et al. about the use of phage display in cancer drug discovery. This methodology, whose efficacy was endorsed in 2018 by the Chemistry Nobel Prize awarded to its inventors, may be applied to find antibodies that specifically bind to the aimed targets. The review shows different approaches to implement the methodology, mainly: types of libraries (mostly in terms of their origin) that may be made; diverse selection modes; and how elution can improve the finding of a good candidate. The manuscript is well written and is scientifically sound, and it deserves publication in Viruses. The authors should however address two issues before publication:
1. An important issue for this reviewer is the ability of antibodies to cross membranes or, more generally speaking, to act intracellularly. This reviewer is concerned about this topic because of its importance regarding the different kinds of molecules that are “druggable”, as small organic molecules, peptides, proteins/antibodies, and so on. The authors should convey at least something about the mechanism the antibodies work against cancer. Do they just work on surface proteins/receptors? Do they cross membranes? Do they cross solid tumors? Crossing membranes/biological barriers is a hot topic in drug discovery, and the authors should consider include explanations about it.
NOTE: the authors mention the example of [line 381] PKM2, a protein overexpressed in lung cancer cells under hypoxia; however the idea is to comment/discuss/include the topic of the location of the targets as a general topic to deal with.
2. A very minor thing is why Table 1 has rows in yellow? The authors should explain it or just remove the color.
The manuscript is very well written to this referee's opinion, very easy to read. So, no complains about it.
Author Response
- An important issue for this reviewer is the ability of antibodies to cross membranes or, more generally speaking, to act intracellularly. This reviewer is concerned about this topic because of its importance regarding the different kinds of molecules that are “druggable”, as small organic molecules, peptides, proteins/antibodies, and so on. The authors should convey at least something about the mechanism the antibodies work against cancer. Do they just work on surface proteins/receptors? Do they cross membranes? Do they cross solid tumors? Crossing membranes/biological barriers is a hot topic in drug discovery, and the authors should consider include explanations about it.
NOTE: the authors mention the example of [line 381] PKM2, a protein overexpressed in lung cancer cells under hypoxia; however the idea is to comment/discuss/include the topic of the location of the targets as a general topic to deal with.
Answer: The reviewer's suggestion is quite pertinent. The purpose of this review is to discuss the different anti-tumor antibodies developed by the phage display tool. In addition to antibodies against receptors, important antibodies act by inhibiting tumor cells intracellularly. We added new paragraphs on intracellular antibodies, including the process of crossing membranes, and we discussed the mechanisms of action of these antibodies, in lines 424-447.
Lines 424-447: “Some therapeutic approaches with anti-tumor antibodies involve mechanisms of intracellular action. Specific binding to cytoplasmic and nuclear molecules that regulate cancer development, improvised delivery of toxins with intracellular antibodies, blocking secretion of tumor receptors on the endoplasmic reticulum, and antibody-mediated alterations in cellular signaling pathways are new forms of immunotherapy in addition to interaction with the cell membrane. Furthermore, intracellular antibodies are an alternative for diagnosis, targeting internal markers. The action of intracellular antibodies faces the difficulty of these molecules crossing the plasmatic membrane and escaping from the pathways of protein degradation. Different strategies have been explored to overcome this challenge. Phage Display is an important tool to allow obtaining smaller antibody formats, such as scFv and VHH, which are more favorable structures for transit through membranes and tissue barriers and for intracellular expression [126, 127].
D'agostino et al. (2022) selected, with a llama immune library, the D11 VHH antibody, specific to the Twist1 factor which inhibits the p53 protein. The activity of D11 VHH was tested in fibrosarcoma cells by intracellular expression of the antibody using lentiviral vectors [128]. Pastushok et al. (2019) developed, using an immunized phage display library, an anti-RAD51 scFv-Fc. RAD51 is a nuclear protein involved in DNA stability and is overexpressed in many types of cancer. For the intracellular action of inhibiting RAD51, the authors fused the recombinant antibody to a cell penetrating peptide, (CPP). CPPs are peptides used for membrane transit of different macromolecules. This type of association of antibodies with CPPs are called TransMabs [129]. Another immunotherapy alternative in Cancer is the construction of bispecific antibodies containing the anti-tumor antibody fused to a fragment of an autoantibody. Autoantibodies are known to have intrinsic capacities to cross cell membranes [130].”
- A very minor thing is why Table 1 has rows in yellow? The authors should explain it or just remove the color.
R: We are grateful for the relevant observation. The idea is to highlight the antibodies approved for use against cancer. We removed the yellow marking and restructured Table 1 to clarify the characteristics of these antibodies among those developed by phage display.
